# The Role of COVID-19 Vaccinal Status in Admitted Children during OMICRON Variant Circulation in Rio de Janeiro, City—Preliminary Report

**DOI:** 10.3390/vaccines10040619

**Published:** 2022-04-15

**Authors:** André Ricardo Araujo da Silva, Bernardo Rodrigues Rosa de Carvalho, Monica del Monaco Esteves, Cristiane Henriques Teixeira, Cristina Vieira Souza

**Affiliations:** 1Materno-Infantil Department, Faculty of Medicine, Fluminense Federal University, Niterói 24033-900, Brazil; bernardorrc@id.uff.br (B.R.R.d.C.); monicaesteves@id.uff.br (M.d.M.E.); 2Infection Control Committee, Prontobaby Group, Rio de Janeiro 20540-100, Brazil; ccih@cplagoa.com.br (C.H.T.); ccih@prontobaby.com.br (C.V.S.)

**Keywords:** COVID-19, children, vaccinal status

## Abstract

Objective: To evaluate COVID-19 vaccination status in admitted children in 2020–2021 and during the OMICRON variant circulation (2022), a period when children older than 12 years of age had received two doses of COVID-19 vaccines. Design: An observational retrospective study. Patients with confirmed COVID-19 were compared in two different periods: 2020–2021 when adolescents aged 12–18 years had not received the complete COVID-19 vaccine, and 2022 when children older than 12 years had received the complete Pfizer-BioNTech vaccine scheme. Setting: Two pediatric hospitals in Rio de Janeiro city. Patients: Children aged < 18 years with confirmed COVID-19. Intervention: None. Main outcome: Vaccination status for COVID-19 on admission. Results: In total, 300 patients were admitted with confirmed COVID-19 (240 in 2020–2021 and 60 in 2022). The distribution of patients according to the age-groups was: 0–2 years (33.3% in 2020–2021 and 53.4% in 2022), 2–5 years (21.7% in 2020–2021 and 10% in 2022), 5–11 years (29.2% in 2020–2021 and 28.3% in 2022), and 12–18 years (15.8% in 2020–2021 and 8.3% in 2022) (*p* = 0.076). The median length of stay was six days in 2020–2021 and six days in 2022 (*p* = 0.423). We verified six deaths in the first analysis period and one death in the second one (*p* = 0.894). Of the 60 children admitted in 2022, 58 (96.7%) did not receive the complete COVID-19 vaccine scheme available. Conclusions: We verified in a “real-world condition” the ability of the Pfizer-BioNTech vaccine to prevent hospitalization in children over 12 years of age.

## 1. Introduction

SARS-CoV-2 is the causative agent of COVID-19, and the first cases of the disease were formally report as the outbreak of pneumonia in 44 patients from Wuhan city, Hubei province, China, in December 2019 [1]. The initial cases were linked to a seafood wholesale market; however, there is evidence of virus circulation outside China since the virus was identified in samples taken at blood banks before December 2019 in European countries and Brazil [2,3]. The disease quickly spread to other countries, especially during the first months of 2020 in Europe, and the World Health Organization (WHO) declared the pandemic status on 11 March 2020 [4].

The COVID-19 pandemic was still ongoing in March 2022, after more than two years of the initial description of SARS-CoV-2 [1]. Since then, the disease has constantly been manifested in all continents and occurred in waves, when a large number of cases were reported in specific countries or regions, contributing to overcrowded of health systems and increasing number of deaths. During the two years of the pandemic, at least five variants of concern spread around the world, with the B.1.1.529 variant (OMICRON) being the most prevalent in most countries, including Brazil, since the first report on 24 November 2021 in South Africa [5].

According to the Brazilian Health Ministry, until the 48th epidemiological week (4 December 2021), 1422 children died in consequence of acute respiratory syndrome distress due to COVID-19, which represents 0.38% of the total deaths (372,954) [6]. Despite the relatively low number of deaths, the absolute number of child deaths from COVID-19 was 8.5 times higher than all other respiratory virus deaths combined, causing the same syndrome [6].

During the COVID-19 pandemic, several initial strategies were employed to reduce the spread of the disease and including non-pharmacologic interventions (NPI), such as restrictions on mass gatherings, stay-at-home orders, restrictions on international travel, testing policy, contact tracing, public information campaigns, shutdown implementation, the public wearing of masks, closing of nonessential businesses, closing borders, and school closing during the lack of appropriate vaccine [7,8]. The NPI, when applied together, was efficient in controlling the spread of disease in the first year of the pandemic; however, the development of specific vaccines was necessary to reduce hospitalization numbers and deaths due to the disease.

Vaccine administration is a key point to modifying the disease impact on the whole population, including children, and its effects were recently evaluated in adolescents and children older than five years of age [9,10]. In Brazil, the first COVID-19 vaccine available was CoronaVac for older persons and specific populations in January 2021. Only in August 2021, BNT162b2 (Pfizer-BioNTech), a national campaign, started for children between 12 and 17 years old. The BNT162b2 (Pfizer-BioNTech) vaccine was administered in two doses 12 weeks apart, and on 17 January 2022, the COVID-19 vaccination for children between five and 11 years old began in Brazil.

As the effect of COVID-19 in children needs to be studied in a “real-world setting” regarding the distribution age of admitted children and outcomes, we aim to evaluate COVID-19 vaccination status in children admitted in two different periods, the second during the OMICRON variant circulation in Rio de Janeiro city and when children older than 12 years of age had received two doses of the COVID-19 vaccines.

## 2. Methods

We conducted an observational retrospective study on children from 0 to 18 years old who were admitted to two pediatric hospitals in Rio de Janeiro city, Brazil. The hospitals receive patients from their emergency departments or those referred to hospitalization (inwards or intensive care units) of other units. Unit 1 is the largest pediatric hospital in Rio de Janeiro city, has a 135-bed capacity, is located in the North area of the county, and receives patients from the metropolitan region of Rio de Janeiro state. Unit 2 is located in the south of the city with a capacity of 45 beds and also receives patients from Rio de Janeiro city and the metropolitan region of Rio de Janeiro state.

Children admitted for more than 24 h between 1 January 2020 and 20 February 2022 in any ward were included if they filled the WHO case definitions [11]. Children admitted but transferred to other hospitals were excluded.

In the city of Rio de Janeiro, children aged from 12 to 17 were only vaccinated (two doses) in mid-December 2021. Children aged from 5 to 11 started receiving the COVID-19 vaccine on 17 January 2022. In Brazil, until this moment, the vaccines BNT162b2 (Pfizer-BioNTech) and CoronaVac are approved for children aged from 5 to 18 and from 6 to 18, respectively. Variant identification of positive cases has not occurred.

The distribution age of confirmed cases was compared with the period before the COVID-19 vaccine availability for children aged between 12 and 18 years (2020 and 2021). Considering that the complete protection for children between 12 and 18 years was obtained only on 30 December 2021, we admit that all children with confirmed COVID-19 were not fully protected against SARS-CoV-2 in 2020 and 2021.

A descriptive analysis of age, gender, length of stay, outcomes, and vaccination status was performed using Microsoft Excel. The Shapiro–Wilk test calculated the sample normality. We used the Chi-square test for categorical variables, the Mann–Whitney U test for continuous variables, and standard residuals where appropriate. A *p*-value lower than 0.05 was considered statistically significant.

The study was submitted and approved by the Ethics Committee of the Faculty of Medicine (Universidade Federal Fluminense) and the Prontobaby Group, under number 4.100.232, dated 20 June 2020. A consent statement for data from the patients or their parents or guardians was required for each child included.

## 3. Results

In total, 300 patients were admitted with confirmed COVID-19 (240 in 2020–2021 and 60 in 2022) (until 20 February). Table 1 shows the demographic data of confirmed patients, length of stay, and outcomes.

In the 2020–2021 period, 128/240 (53.3%) patients were admitted to pediatric intensive care units (PICUs), while 112/240 (46.7%) were admitted to wards. In 2022, 31/60 (51.6%) patients were admitted to PICUs and 29/31 (48.3%) were admitted to wards (*p* = 0.817).

Five patients older than 12 years were admitted in 2022. Two of these patients received two doses, two received one dose, and one received zero. All patients admitted in the 2020–2021 year were not fully immunized.

The media numbers of admitted cases per/week in 2020–2021 and 2022 were 2.55 and 10, respectively (*p* = 0.226).

## 4. Discussion

Since the COVID-19 vaccines had their use authorized for children and adolescents older than 12 years, it is necessary to evaluate the real impact in this population regarding the number of hospitalized children that received two doses of Pfizer-BioNTech and deaths. Despite the significant advances in controlling the disease, the end of the COVID-19 pandemic was not declared by 11 March 2022 (two years after the beginning). In this report, we evaluated COVID-19 vaccination status in children admitted in two different periods, the second one during the OMICRON variant circulation and a period of time when children older than 12 years of age had received two doses of the COVID-19 vaccines.

When analyzing the distribution age of the studied children, we verified the same pattern of distribution age of patients admitted in the first period (2020–2021) described by Hillesheim and colleagues among hospitalized children and adolescents who were diagnosed with severe acute respiratory syndrome (SARS) due to COVID-19 in Brazil in 2020 [12]. Swann and colleagues also verified the homogeneous distribution age in a study involving 651 children and young people aged less than 19 years admitted to 138 hospitals in England, Wales, and Scotland between 17 January and 3 July 2020. In this report, the age groups of children younger than one year represented 34.5% of admitted children, 16.6% were 1–4 years old, 14.1% were 5–9 years old, 14.4% were 10–14 years old, and 20.3% belonged to the 15–19-year-old group [13].

Comparing the distribution age of patients admitted in the second analysis period, we verified more admitted children between 0 and 2 years during the OMICRON season and fewer children older than 12 years compared with the previous two years. Although we did not find statistical significance between the distribution ages of admitted children, few children older than 12 years hospitalized during the OMICRON season in Rio de Janeiro could represent a positive effect of vaccination in this age group that started in the 2021 year.

Children were less affected than adults, and this finding was verified in all continents. A meta-analysis conducted by Bhuiyan and colleagues analyzing studies conducted in the first six months of the pandemic suggests that more than 90% of children developed mild to moderate disease, and only 7% were severe cases requiring intensive care unit (ICU) or high-dependency care unit (HDU) [14]. Considering that the disease occurred in different waves over time, one concern is to identify different severity patterns in children. Krishnamurthy et al. described the same presentation of the disease in children when analyzing two different periods (March 2020–Jan 2021 and Feb 2021–July 2021) in India [15]. In this report, 74.1% were mildly symptomatic in the first period versus 80.2% in the second period.

In our study, the severity pattern of COVID-19 was the same when both periods were compared, without statistical significance, with more than half admitted to pediatric intensive care units. This finding is different from that reported by Götzinger et al., who presented data from 582 admitted children in 82 participating healthcare institutions across 25 European countries, and only 48/582 (8.3%) were admitted to intensive care units [16]. In a study conducted in five Latin American countries, 52/409 (12.7%) children with COVID-19 were admitted to PICUs [17]. This difference can be explained given that the admissions to our intensive care units were recommended for all children that need supplementary oxygen of any type (catheter, non-invasive ventilation, or invasive ventilation).

Although a higher number of patients were admitted to PICUs, the length of stay was the same when both periods (6 days) were compared, showing favorable outcomes for children, even considering the circulation of different COVID-19 variants. The results were similar to those reported by Garcia-Salido and colleagues in 74 children admitted to PICUs in Spain. The length of stay in patients with the multisystem inflammatory syndrome (MIS-C) was five days (2.5–8 days) versus 6.5 days (3.3–10.8 days) in the group without MIS-C [18].

In the first six weeks of 2022, Rio de Janeiro city and Brazil experienced a higher circulation of the OMICRON variant, causing an increase in the number of cases of all ages, including children [19]. Between March 2020 and December 2021 (the first analysis period of admitted children), the most predominant variants were Gamma (until July 2021) and Delta (after July 2021 until December 2021) [20]. Confirming this finding, the average number of patients/week admitted with COVID-19 in 2022 was almost four times higher than the previous year, representing an exponential increase in cases in Rio de Janeiro due to the OMICRON circulation.

Of the 60 children admitted in 2022, only 2 (3.3%) received the complete COVID-19 vaccine available, showing that almost all admitted cases of the disease occurred in children where the vaccine was not available for their respective ages or did not receive two doses. The COVID-19 vaccines were only administrated in children between 5 and 12 years as of mid-January 2022. Recently, a similar “real world” study of 464 hospitalized US children and adolescents aged 12–18 years (179 case patients and 285 controls) with a mean age of 15 years, 72% with at least one underlying condition, including obesity, verified that the effectiveness of two doses of the Pfizer-BioNTech vaccine against COVID-19 hospitalization was 93% (95% CI = 83–97%) during the period of variant B.1.617.2 (Delta) prevalence [21].

Despite the promising data about COVID-19 vaccine efficacy, several countries are suffering from a massive campaign on social media with vaccine hesitancy and the anti-vaccine movement [22]. In a cross-sectional survey conducted in Turkey, 506 pediatricians were interviewed, and a considerable number of physicians do not recommend COVID-19 vaccines to their children and patients [23].

Since the beginning of the COVID-19 pandemic, more than 450 million cases and 6 million deaths have occurred worldwide [24]. Until 10 March 2022, Brazil was the second country in the absolute number of deaths, with more than 652,829 attributed to the SARS-CoV-2 [24].

We verified that mortality due to COVID-19 was similar in the two periods studied; however, the true impact of COVID-19 vaccines in reducing mortality in children needs to be accessed in prospective studies, evaluating patients after vaccination for different children ages. Our mortality rates were similar to other reports in both analysis periods. For example, a systematic review of the multisystem inflammatory syndrome in children related to COVID-19 described 953 cases mostly characterized by fever, gastrointestinal, and cardiocirculatory manifestations, as well as increased inflammatory biomarkers, with a mortality rate of 1.9% [25]. The relatively low number of deaths due to COVID-19 was also verified by analysis in seven countries (USA, UK, Italy, Germany, Spain, France, and South Korea), where the death rate found was 0.17 per 100,000 population, comprising 0.48% of the estimated total mortality from all causes in a regular year [26]. In a study conducted when the COVID-19 vaccine was available, Zambrano et al. reported the effectiveness of two doses of Pfizer-BioNTech vaccine received ≥28 days before hospital admission in preventing MIS-C. Patients aged 12–18 years hospitalized in 24 pediatric hospitals in 20 states, from July 1 to 9 December 2021, were evaluated. This period corresponded to the time most MIS-C patients could temporally relate to the SARS-CoV-2 B.1.617.2 (Delta) variant predominance [27]. The results estimated an effectiveness of 91% (95% CI = 78–97%) for two doses of Pfizer-BioNTech vaccine against MIS-C [27].

Besides the vaccination, other factors such as the virulence of SARS-CoV-2 variants should be of concern in this condition. A recent report on incidence rates and clinical outcomes of SARS-CoV-2 infections in children younger than five years in the US verified that severe clinical outcomes were less frequent with the OMICRON variant than with the Delta variant [28].

Our study has some limitations. The first was that the data were from only two pediatric hospitals in the city. Nevertheless, the data could be representative because both hospitals receive children from all neighborhoods in the city and metropolitan areas throughout the state. In addition, the increase in the number of admitted cases coincided with that described in Rio de Janeiro city. The second limitation was not identifying the confirmed COVID-19 variant due to the limiting resources available in the units. However, the Rio de Janeiro state monitoring panel showed that the OMICRON variant became predominant in the city as of December 2021 [20]. Finally, our report intends to be a long-term preliminary report following the necessity to confirm positive findings of vaccine effects on reducing hospitalization.

## 5. Conclusions

In conclusion, we verified the ability of the Pfizer-BioNTech vaccine to avoid hospitalization in children older than 12 years of age in a “real-world-condition”. Children under 12 years need to be followed to verify COVID-19 impact and to reduce hospitalization and deaths in this age group.

## Figures and Tables

**Table 1 vaccines-10-00619-t001:** Demographic data of admitted cases in two pediatric hospitals (Rio de Janeiro city, 1 January 2022–20 February 2022).

	2020–2021 N = 240	2022 N = 60	*p* Value
Gender	104 (43.3) 136 (56.7)	30 (50) 30 (50)	0.352
-Female			
-Male			
Age	80 (33.3) 52 (21.7) 70 (29.2) 38 (15.8)	32 (53.4) 6 (10) 17 (28.3) 5 (8.3)	0.076
0–2 years			
2–5 years			
5–11 years			
12–18 years			
Length of stay (MEDIAN in days)	6 (1–164)	6 (1–28)	0.423
Outcome after 7 days * -Discharged -Death	234 (97.5) 6 (2.5)	57 (95) 1 (1.7)	0.894

* Two patients still hospitalized by 20 February 2022.

## Data Availability

Not applicable.

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
