# Peer review of "The Role of COVID-19 Vaccinal Status in Admitted Children during OMICRON Variant Circulation in Rio de Janeiro, City—Preliminary Report"

_vaccines, 2022, doi:10.3390/vaccines10040619_

Round 1
Reviewer 1 Report
In this manuscript, the authors tried to evaluate COVID-19 vaccine (Pfizer-BioNTech) protection against hospitalization in children older than 12 years.
The study theme is critical since we need more vaccine data support for children and teens. However, this study itself did not provide clear enough information to give us indications.
Firstly, the sample amount is limited. Only 38 (2020/2021) plus 5 (2022) patients were 12-18 years old and allowed to get the vaccination. The authors mixed all ages of admitted children to import more parameters to be considered.
In addition, the hospitalization date range among the samples is too broad. The authors compared the patients between 2020/2021 and early 2022, the years that the circulating SARS-CoV-2 strains were very different.
Besides the vaccination, other factors, such as the virulence of SARS-CoV-2 variants, should be considered in this condition.
Taken together, I don’t think the investigated samples in this manuscript were suitable for evaluating COVID-19 vaccination status.
Author Response
The study theme is critical since we need more vaccine data support for children and teens. However, this study itself did not provide clear enough information to give us indications.
Firstly, the sample amount is limited. Only 38 (2020/2021) plus 5 (2022) patients were 12-18 years old and allowed to get the vaccination. Thank you for your observation. After two years of COVID-19 it’s well described that COVID-19 is a mild-moderate illness in children compared with adults with low rates of admission and morality when compared with adults ( Bhopal SS, Bagaria J, Olabi B, Bhopal R. Children and young people remain at low risk of COVID-19 mortality. Lancet Child Adolesc Health. 2021 May;5(5):e12-e13). For this reason, even after two years of pandemic, number of admitted children remains low, as we described in out manuscript. In lines 93-95 we inform that all chidren of 2020/2021 period were considered not fully protected.
The authors mixed all ages of admitted children to import more parameters to be considered. The distribution age of children was presented as demographic data comparing two different periods in order to show that distribution age was different during the second period of analysis ( higher relative percentual of children under 2 years and fewer children between 12-18 years in second period compared with the first one) although statistical significance The difference between age groups were calculated by standard residuals ( lines 97-99) and presented in discussion section ( lines 140-146)
In addition, the hospitalization date range among the samples is too broad. The authors compared the patients between 2020/2021 and early 2022, the years that the circulating SARS-CoV-2 strains were very different. We agree with the reviewer and included an explanation in the discussion to clarify this question ( lines 176-178)
Besides the vaccination, other factors, such as the virulence of SARS-CoV-2 variants, should be considered in this condition. We agree with the reviewer. We also access this topic in the discussion section (lines 219-223). A recent article published at JAMA pediatrics also analysed this topic and it was included as reference ( Wang L, Berger NA, Kaelber DC, Davis PB, Volkow ND, Xu R. Incidence Rates and Clinical Outcomes of SARS-CoV-2 Infection With the Omicron and Delta Variants in Children Younger Than 5 Years in the US. JAMA Pediatr. 2022 Apr 1. doi:10.1001/jamapediatrics.2022.0945. Epub ahead of print.)
Taken together, I don’t think the investigated samples in this manuscript were suitable for evaluating COVID-19 vaccination status. As described in the title of article, this is a preliminary report and we inform in our manuscript that a long-term follow is necessary to confirm our findings ( pag 5, lines 231-233)
Reviewer 2 Report
This manuscript is an interesting article because it allows us to see the impact of vaccination campaigns on hospital pressure in pediatrics
Question. In line 46, the authors refer to the epidemiological week of December 4. this seems to indicate that it ends on Saturday. in many countries, it ends on Sunday, so I don't know if the reference to December 4 is correct or if it is an error.
The results section is very concise in relation to other sections such as the introduction and discussion.Los autores deberían comentar los resultados de la tabla 1 especialmente las diferencias en los grupos de edad en los dos períodos.
It would not be enough to comment that there are significant differences between the two periods in the age groups, but we would have to say between which groups the difference occurs. For this, we would have to calculate standardized adjusted residuals. These residuals would tell us between which groups the difference occurs.
The calculations can be done with commercial programs such as IBMSPSS or by calculating the formula manually in Excel. The formula can be found in the following link
"The mean of the number of admitted cases per/week in 2020/2021 and 2022 were 2.6 and 117 10, respectively" it is necessary to know if these differences are statistically significant. The standard deviation of the number of admissions per week should be provided. Finally, a significance test should be performed: a Student's t-test or its nonparametric equivalent Mann Witney U-test.
Author Response
This manuscript is an interesting article because it allows us to see the impact of vaccination campaigns on hospital pressure in pediatrics. Thank for your comments
Question. In line 46, the authors refer to the epidemiological week of December 4. this seems to indicate that it ends on Saturday. in many countries, it ends on Sunday, so I don't know if the reference to December 4 is correct or if it is an error. Thank you for comment. In Brazil, December 4, 2021 was the last day of Epidemiological week 48. The epidemiological week started in December 5, 2021. ( http://portalsinan.saude.gov.br/calendario-epidemiologico-2020/43-institucional/171-calendario-epidemiologico-2021)
The results section is very concise in relation to other sections such as the introduction and discussion.Los autores deberían comentar los resultados de la tabla 1 especialmente las diferencias en los grupos de edad en los dos períodos. Thank you for you comment, we improved the discussion, especially between lines 140 and 146 to access this topic
It would not be enough to comment that there are significant differences between the two periods in the age groups, but we would have to say between which groups the difference occurs. For this, we would have to calculate standardized adjusted residuals. These residuals would tell us between which groups the difference occurs.
The calculations can be done with commercial programs such as IBMSPSS or by calculating the formula manually in Excel. The formula can be found in the following link. Thank you for suggestion. We used this test ( please see lines 97-99)
"The mean of the number of admitted cases per/week in 2020/2021 and 2022 were 2.6 and 117 10, respectively" it is necessary to know if these differences are statistically significant. The standard deviation of the number of admissions per week should be provided. Finally, a significance test should be performed: a Student's t-test or its nonparametric equivalent Mann Witney U-test.Thank you for comment . We calculated this difference according your suggestion. Please see the results in results section ,line 118-119
Round 2
Reviewer 1 Report
I am satisfied with the current version.
Reviewer 2 Report
The authors have answered all the questions that were formulated in the first review.